# Prospective evaluation of deep learning image reconstruction for Lung-RADS and automatic nodule volumetry on ultralow-dose chest CT

**Seung-Jin Yoo**[1], **Young Sik Park**[2], **Hyewon Choi**[3], **Da Som Kim**[4], **Jin Mo Goo**[5], **Soon Ho Yoon**[5]*

**1** Department of Radiology, Hanyang University Medical Center, Hanyang University College of Medicine, Seoul, Republic of Korea, **2** Department of Internal Medicine, Seoul National University Hospital, Seoul National College of Medicine, Seoul, Korea, **3** Department of Radiology, Chung-Ang University Hospital, Chung-Ang University College of Medicine, Seoul, South Korea, **4** Departments of Radiology, Busan Paik Hospital, Inje University College of Medicine, Busan, Korea, **5** Department of radiology, Seoul National University Hospital, Seoul National College of Medicine, Seoul, Korea

* yshoka@snu.ac.kr

**Data Availability Statement:** All relevant data except for chest CT images are within the paper. Chest CT image data cannot be shared publicly

## Abstract

### Purpose

To prospectively evaluate whether Lung-RADS classification and volumetric nodule assessment were feasible with ultralow-dose (ULD) chest CT scans with deep learning image reconstruction (DLIR).

### Methods

The institutional review board approved this prospective study. This study included 40 patients (mean age, 66±12 years; 21 women). Participants sequentially underwent LDCT and ULDCT (CTDIvol, 0.96±0.15 mGy and 0.12±0.01 mGy) scans reconstructed with the adaptive statistical iterative reconstruction-V 50% (ASIR-$V_{50}$) and DLIR. CT image quality was compared subjectively and objectively. The pulmonary nodules were assessed visually by two readers using the Lung-RADS 1.1 and automatically using a computerized assisted tool.

### Results

DLIR provided a significantly higher signal-to-noise ratio for LDCT and ULDCT images than ASIR-$V_{50}$ (all P < .001). In general, DLIR showed superior subjective image quality for ULDCT images (P < .001) and comparable quality for LDCT images compared to ASIR-$V_{50}$ (P = .01–1). The per-nodule sensitivities of observers for Lung-RADS category 3–4 nodules were 70.6–88.2% and 64.7–82.4% for DLIR-LDCT and DLIR-ULDCT images (P = 1) and categories were mostly concordant within observers. The per-nodule sensitivities of the computer-assisted detection for nodules ≥4 mm were 72.1% and 67.4% on DLIR-LDCT and ULDCT images (P = .50). The 95% limits of agreement for nodule volume differences between DLIR-LDCT and ULDCT images (-85.6 to 78.7 mm³) was similar to the within-scan nodule volume differences between DLIR- and ASIR-$V_{50}$-LDCT images (-63.9 to 78.5

because of the private health information policies of participating institutions. Chest CT image data can be shared to researchers who meet the criteria for access to confidential data upon request (contact to: yshoka@snu.ac.kr). The non-author contact information (phone/email/hyperlink) is as follows: phone, 82-2-2072-2266; email, irb@snu.ac.rk; hyperlink, http://hrpp.snuh.org/irb/introirb/_/singlecont/view.do.

**Funding:** This work was supported by GE Healthcare (Grant number: 06-2020-0300). No author received funding in the form of salary from GE Healthcare. URLs to sponsors' websites is: https://www.gehealthcare.co.kr/. The funder had no role in study design, data collection and analysis, decision to publish, or preparation of the manuscript.

**Competing interests:** This work was supported by GE Healthcare (Grant number: 06-2020-0300). There are no patents, products in development, or marketed products associated with this research to declare. This does not alter our adherence to PLOS ONE policies on sharing data and materials.

$mm^3$), with volume differences smaller than 25% in 88.5% and 92.3% of nodules, respectively (P = .65).

## Conclusion

DLIR enabled comparable Lung-RADS and volumetric nodule assessments on ULDCT images to LDCT images.

## Introduction

Large-scale randomized controlled trials have proven that lung cancer screening using low-dose chest CT (LDCT) scans reduces lung cancer mortality by 20–33% in high-risk groups [1,2]. Several countries have implemented LDCT-based lung cancer screening programs, increasing the use of LDCT scans, which are assessed in terms of the Lung Imaging Reporting and Data System (Lung-RADS) categorization or nodule volumetry [3,4]. LDCT scans are also widely used to evaluate various lung diseases since LDCT is easily accessible while providing similar diagnostic performance and less radiation exposure compared to standard-dose CT scans [5]. A recent study by Sakane et al. reported no damage of human DNA from a single low-dose CT scan [6]. However, lung cancer CT screening and pulmonary disease evaluations may often involve the frequent use of follow-up CT scans [7], and the safety of the cumulative radiation exposure remains underexplored.

Ultra-low-dose chest CT (ULDCT) scans are a potential option to reduce radiation exposure. ULDCT scans of the thorax deliver a lower effective dose (0.20–0.49 mSv) than LDCT scans (about 3 mSv) [8–11]. However, ULDCT scans are inevitably accompanied by higher image noise than LDCT scans, hampering image interpretation. Various image reconstruction methods, including model-based iterative reconstruction, have been attempted for ULDCT scans to mitigate the image noise compromise. Recently, a deep learning image reconstruction (DLIR) system was commercialized by GE (TrueFidelity, GE Healthcare), based on a vast amount of training data of standard-dose phantom and patient CT images reconstructed by filtered back projection [12]. Indeed, a few studies have reported that DLIR could provide better image quality of thoracic LDCT and ULDCT scans than the pre-existing iterative reconstruction method [13–15]. Nevertheless, whether DLIR enables Lung-RADS classification and volumetric evaluation of pulmonary nodules in ULDCT scans remains unknown.

Our study aimed to prospectively evaluate whether Lung-RADS and volumetric nodule assessments were feasible in DLIR-ULD CT scans.

## Materials and methods

The Institutional Review Board of Seoul National University Hospital approved this prospective study, and medical staff obtained written informed consent from all participants. The study protocol was registered at the Clinical Research Information Service (CRIS, Registration Number: KCT0004692).

### Study population

The inclusion criteria were adults who visited the outpatient respiratory clinic for the first time and planned to undergo a chest CT scan at a single tertiary hospital from February 2020 through July 2020. We excluded patients with a) a body mass index over 30 $kg/m^2$, b) an

inability to hold their breath sufficiently for chest CT scanning, c) a previous history of procedures or operations that may affect the image quality of CT scans (e.g., central venous catheter, defibrillator implantation, valve replacement) and d) contraindications for CT scans (e.g., pregnancy). 40 patients (mean age ± standard deviation, 66±12; 21 women) were included in this study.

## CT acquisition

Consecutive full-inspiratory thoracic LDCT scans at 120 kVp and ULDCT at 100 kVp were acquired in all patients using a single CT machine (Revolution CT; GE Healthcare, Waukesha, WI, USA) and the scanning parameters were as follows: tube voltage, 120 kVp with a mAs of 25 for LDCT, 100 kVp with a mAs of 5 for ULDCT; automatic tube current modulation; gantry rotation time, 280 ms; detector configuration, 128 × 0.625 mm; beam pitch, 1.53; matrix, 512x512; reconstruction increment and section thickness, 1.25 mm; noise index for LDCT, 28 and noise index for ULDCT, 33. Each CT scan was reconstructed with adaptive statistical iterative reconstruction V with 50% blending with FBP (ASIR-V$_{50}$), considering the results of previous studies with various ASIR-V blending levels [16,17], and DLIR-high level (DLIR-H, TrueFidelity). Thus, four CT image series were generated per patient, resulting in 160 CT series.

## Objective and subjective image quality assessment

The objective CT image quality was calculated using the signal-to-noise ratio (SNR) on a picture archiving and communication system (PACS). Two thoracic radiologists (S.H.Y and S.J. Y.; 16 and 7 years of experience in chest CT interpretation) independently drew six circular regions of interest (ROIs) in the lung parenchyma while avoiding vessels or visible pathology on DLIR-LDCT and DLIR-ULDCT images. The ROIs were placed in bilateral upper, mid, and lower lungs with similar sizes of 50–100 mm$^2$. Then, the ROIs were copied and pasted into the LDCT and ULDCT with ASIR-V50 images. The mean and standard deviation of the ROIs in Hounsfield units (HU) were recorded and represented the signal and noise, respectively. The SNR was calculated as follows: SNR = |Mean HU$_{ROI}$| / SD$_{ROI}$.

Two board-certified thoracic radiologists (H.C. and D.S.K.; 7 years of experience in chest CT interpretation) visually examined the paired LDCT and ULDCT images reconstructed with ASIR-V$_{50}$ and DLIR-H in a random order. The captured CT images were prepared at the aortic arch level, right middle lobar bronchus level, and right diaphragm level. The pulmonary lesion images were also provided if they existed. The CT images were captured in the lung window setting (window level, -550 HU; window width, 1700 HU), and great vessel–level images were also provided in the mediastinal window setting (window level, 45 HU; window width, 450 HU). The radiologists subjectively scored the image quality regarding image noise, visibility of small structures, lesion conspicuity, and diagnostic acceptability (S1 Table). They were blinded to the radiation dose information and reconstruction algorithm.

## Nodule detection and Lung-RADS evaluation by readers

One board-certified thoracic radiologist (S.J.Y.) reviewed 40 DLIR-LDCT scans thoroughly and detected nodules of Lung-RADS 1.1 category 3 or over for gold standard. The latter two thoracic radiologists (H.C. and D.S.K.) independently reviewed pairs of 40 DLIR-LDCT and 40 DLIR-ULDCT series on the PACS using Lung-RADS 1.1 in random order. The radiologists detected nodules of Lung-RADS 1.1 category 3 or over as follows: solid nodules with a mean diameter ≥6 mm, part-solid nodules with a total diameter ≥6 mm with a solid component <6 mm, nonsolid nodules with a diameter ≥30 mm.

## Volumetric nodule evaluation

Nodule volumetry was conducted using the vendor-provided Lung VCAR (volume computerized assisted reporting, version 14.0–8.11, GE Healthcare), which is an automatic CT lung nodule detection tool designed to detect and quantify nodules to assist the readers. Two radiologists (S.H.Y. and S.J.Y.) prepared gold-standard nodules in consensus with a maximal axial diameter ≥4 mm and recorded the CT characteristics of the nodules (solid, subsolid, calcified). Twelve patients' CT scans were excluded due to multiple lung nodules (more than 20) due to pulmonary tuberculosis (n = 7), non-tuberculous lung disease (n = 4), and hematolymphangitic metastasis (n = 1). Of the remaining 28 patients, eight patients had no lung nodules ≥4 mm in diameter. As a result, 43 lung nodules (solid, 18; subsolid, 7; calcified, 18) from 20 patients were included in the VCAR evaluation (Fig 1). VCAR produced the maximum x-axis diameter and volume of the nodules (Fig 2). The Lung-RADS 1.1 category by volume of each nodule was also recorded.

To evaluate the reliability of Lung VCAR's nodule detection and volume measurement ability, we additionally applied LuCAS (Monitor Corporation, Seoul, Korea), a deep-learning based CAD for nodule detection and segmentation, on LDCT scans with ASIR-V$_{50}$ reconstruction and compared the nodule detection sensitivity and measured volume of the 43 lung nodules.

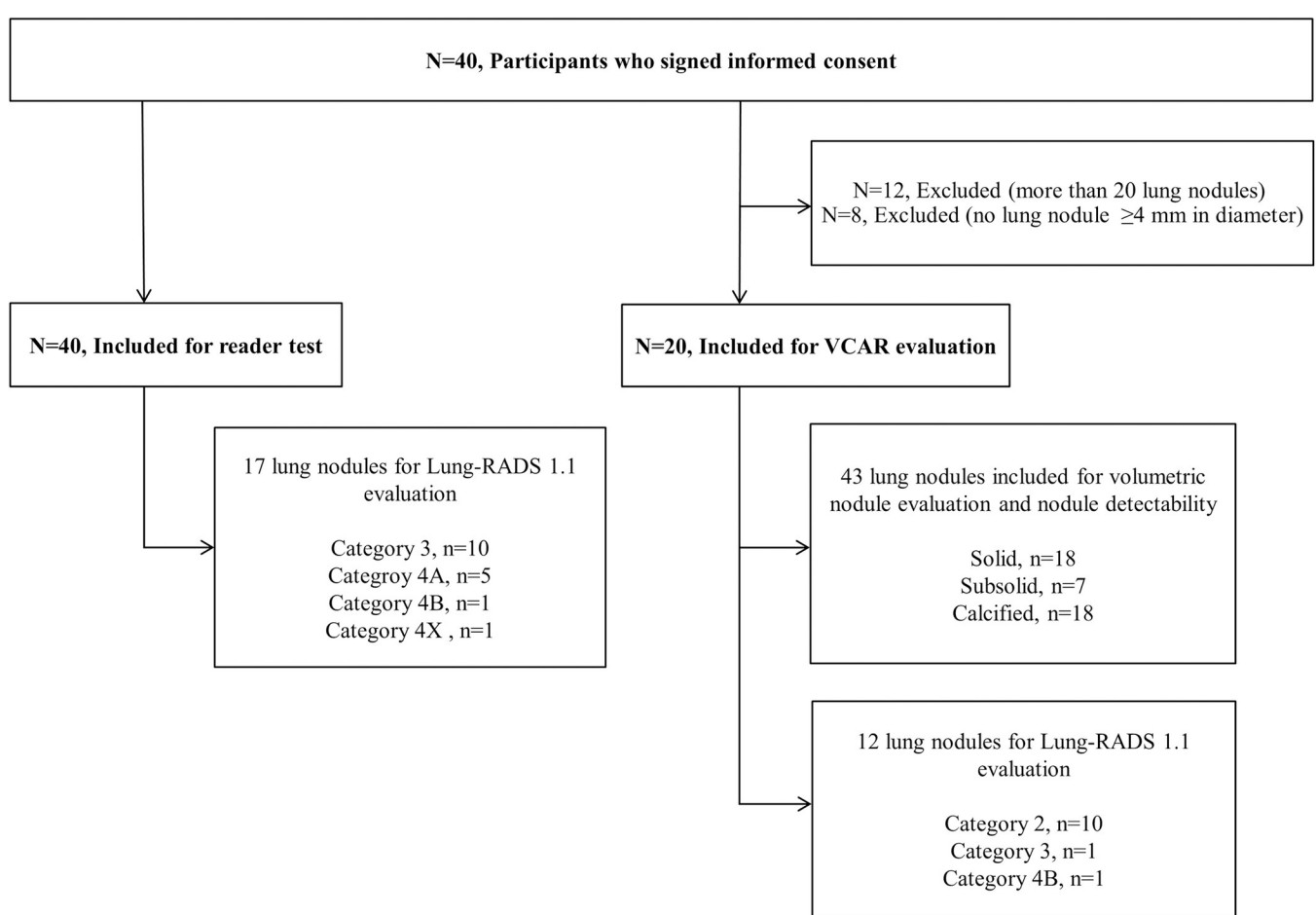

**Fig 1. Flowchart of participants and lung nodules included for the nodule detectability and Lung-RADS category evaluation.** VCAR = volume computerized assisted reporting.

(A)  (B)

(C)  (D)  (E)

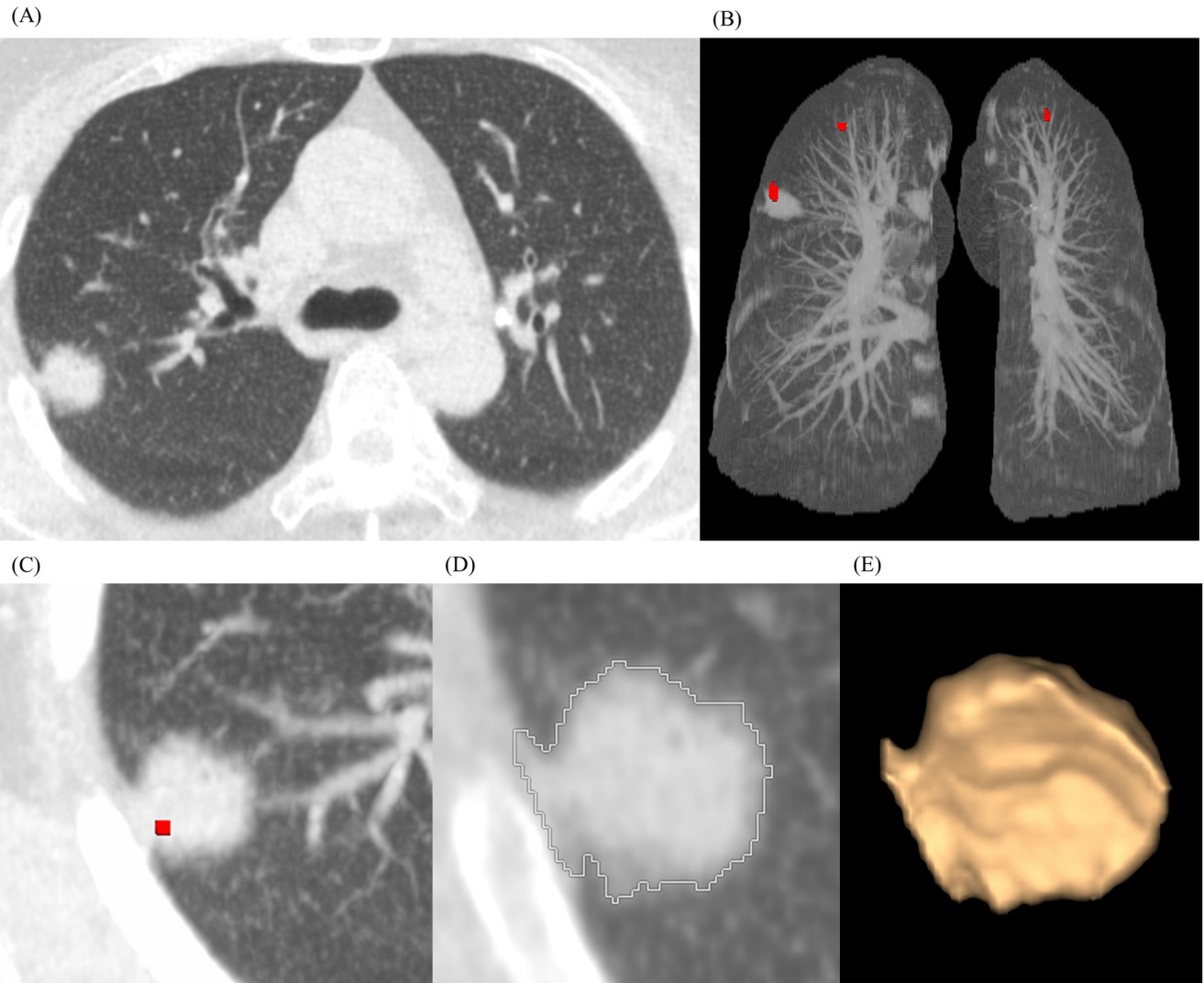

**Fig 2. Detection and volumetric measurement of a lung nodule by VCAR.** Ultralow-dose chest CT scan with deep learning image reconstruction-high level of a 60-year-old female patient with a 1.7-cm solid nodule with a spiculated margin at the right upper lobe (A). Lung volume computerized assisted reporting (VCAR) software automatically detected the right upper lobe nodule and depicted it as a red dot on three-dimensional air maximal intensity projection view (B) and axial maximum intensity projection view (C). The nodule was segmented (D), reconstructed as three-dimensional volume rendering image (E), and the x-, y-, and z-axis diameter, volume, and mean attenuation of the nodule were calculated.

## Statistical analysis

The objective data are expressed as means ± standard deviations. For objective and subjective image quality assessment, the LDCT and ULDCT scans—each with two reconstruction algorithms—were compared using repeated-measures analysis of variance and post-hoc pairwise comparisons after Bonferroni correction. A Bonferroni-corrected p value of <0.008 (0.05/6) was considered to indicate statistical significance. Additionally, for the subjective image quality assessment, interobserver agreement was calculated using intraclass correlation coefficients (ICCs) with a two-way model on each quality assessment subject. An ICC of 0.76–1.0 indicates excellent agreement; 0.40–0.75, fair to good agreement; < 0.40, poor agreement [18]. Reader agreement between LDCT and ULDCT for the Lung-RADS 1.1 nodule classification was evaluated using Cohen's kappa: < 0.21, slight agreement; 0.21–0.40, fair agreement; 0.41–0.60,

**Table 1. Characteristics of study participants.**

| Variables | All Participants (n = 40) |
|---|---|
| Age (y) * | 66±12 |
| Sex † | |
| Women | 21 (53) |
| Men | 19 (47) |
| Height (cm)* | 161.5±8.0 |
| Weight (kg)* | 62.0±11.2 |
| Body mass index (kg/m2)* | 23.7±3.3 |

* Data are means ± standard deviations.

† Data are number of patients with percentage in parentheses.

moderate agreement; 0.61–0.80, substantial agreement; 0.81–1.00, almost perfect agreement [19,20]. The McNemar test was conducted to compare the nodule detection sensitivities of each radiologist and VCAR between DLIR-LDCT and DLIR-ULDCT and to compare the nodule detection sensitivity of VCAR and LuCAS. Bland-Altman plots were used to evaluate differences in nodule volume and diameters among ASIR-V$_{50}$-LDCT, DLIR-LDCT, and DLIR-ULDCT, and the chi-square statistic was used to compare the proportion of nodules with changes of volume more than 25% or diameter more than 1.5 mm. All statistical analyses were performed using MedCalc software version 20.011 (MedCalc Software Ltd, Belgium).

## Results

### Demographics

Table 1 shows the characteristics of the study participants. The mean and standard deviation of body mass index was 23.7±3.3 kg/m$^2$ (Table 1). The mean and standard deviation of the CTDIvol and DLP values were 0.96±0.15 mGy and 40.53±6.04 mGy·cm for the LDCT scans and 0.12±0.01 mGy and 5.09±0.40 mGy·cm for the ULDCT scans, respectively. When a conversion factor of 0.014 mSv/mGy·cm was applied, the mean effective doses were 0.57±0.08 mSv and 0.07±0.01 mSv for LDCT and ULDCT, respectively [21].

### Objective and subjective image quality assessment

The results of the objective image quality assessment are presented in Table 2. DLIR-H provided significantly lower image noise and higher SNR in both LDCT and ULDCT scans than ASIR-V$_{50}$ (all P < .001). DLIR-LDCT showed significantly lower noise and significantly higher SNR than DLIR-ULDCT (both P < .001).

Table 2 also summarizes the results of the subjective image quality assessment. In both LDCT and ULDCT, DLIR-H yielded better results in terms of image noise, small structure visibility, lesion conspicuity, and diagnostic acceptability than ASIR-V$_{50}$ (Figs 3 and 4). Nevertheless, regarding LDCT, image noise (P = .01), visibility of small structures (P = 1), lesion conspicuity (P = 1), and diagnostic acceptability (P = .18) did not significantly differ between ASIR-V$_{50}$ and DLIR-H. In the ULDCT scans, the image noise, visibility of small structures, and diagnostic acceptability were significantly superior in DLIR-H compared to ASIR-V$_{50}$ (all P < .001). When comparing LDCT and ULDCT scans reconstructed by DLIR-H, LDCT showed significantly superior results in all categories (all P < .001). The ICC for interobserver agreement was excellent for image noise and diagnostic acceptability (0.82 and 0.86,

**Table 2. Results of objective and subjective image quality assessment.**

| | LDCT | | ULDCT | | | P-value of paired comparisons* | | |
| | ASIR$_{V50}$ | DLIR-H | ASIR$_{V50}$ | DLIR-H | p-value | ASIR$_{V50}$-LDCT vs. DLIR-LDCT | ASIR$_{V50}$-ULDCT vs. DLIR-ULDCT | DLIR-LDCT vs. DLIR-ULDCT |
|---|---|---|---|---|---|---|---|---|
| **Objective analysis** | | | | | | | | |
| **Signal (HU)** | -887.9 ±38.6 | -885.11 ±38.4 | -860.3 ±93.1 | -872.12 ±47.7 | < .001 | < .001 | .01 | < .001 |
| **Noise (HU)** | 26.9±12.5 | 18.9±12.7 | 49.1±42.3 | 30.8±4.8 | < .001 | < .001 | < .001 | < .001 |
| **Signal-to-noise ratio** | 34.8±7.1 | 51.2±13.1 | 21.5±6.4 | 29.1±5.4 | < .001 | < .001 | < .001 | < .001 |
| **Subjective analysis** | | | | | | | | |
| **Image noise** | 4.6±0.5 | 5.0±0.2 | 3.0±0.9 | 3.7±0.8 | < .001 | .01 | < .001 | < .001 |
| **Visibility of small structures** | 4.6±0.9 | 4.7±0.9 | 2.4±1.1 | 3.3±0.9 | < .001 | 1 | < .001 | < .001 |
| **Lesion conspicuity** | 4.7±0.7 | 4.9±0.2 | 2.8±1.3 | 3.3±1.2 | < .001 | 1 | .14 | < .001 |
| **Diagnostic acceptability** | 4.8±0.4 | 5.0±0.0 | 3.3±0.74 | 3.8±0.8 | < .001 | .18 | < .001 | < .001 |

Data are mean ± standard deviation.

* Bonferroni corrected p value < .008 indicated a statistically significant difference.

LDCT = low-dose chest computed tomography, ULDCT = ultralow-dose chest computed tomography, ASIR-V = adaptive statistical iterative reconstruction-V,

DLIR-H = deep-learning image reconstruction-high level, HU = Hounsfield units.

respectively) and fair to good for visibility of small structures and lesion conspicuity (0.67 and 0.73, respectively).

## Nodule detectability and Lung-RADS categorization by readers

The radiologists' sensitivity for nodule detection is summarized in Table 3. Of the 80 CT scans, which included 40 patients' LDCT and ULDCT scans with DLIR-H reconstruction, 17 nodules belonged to Lung-RADS 1.1 category 3 or over: 10 category 3 nodules, five category 4A nodules, one category 4B nodule, and one category 4X nodule (Fig 1). Radiologists' nodule detection sensitivities did not significantly differ between DLIR-LDCT and DLIR-ULDCT (Reader 1, P = 1; Reader 2, P = 1).

In terms of Lung-RADS 1.1 categories, the two radiologists did not detect or down-categorized three and two nodules each in DLIR-ULDCT compared to in DLIR-LDCT (S2 Table). Two nodules were up categorized from category 4A to 4B in DLIR-ULDCT compared to DLIR-LDCT by reader 2 (S2 Table). The reader agreement between LDCT and ULDCT was 0.73–0.76 (substantial agreement).

## Nodule detectability and Lung-RADS categorization by VCAR

The sensitivity of nodule detection of VCAR is summarized in Table 4. The nodule detection sensitivity by VCAR was equivalent for DLIR-H and ASIR-V$_{50}$ reconstructions in LDCT (ASIR-V$_{50}$ vs. DLIR-H, 72.1% vs. 72.1%, P = 1). In the ULDCT scans, the nodule detection sensitivity of DLIR-H was comparable to that of ASIR-V$_{50}$, (ASIR-V$_{50}$ vs. DLIR-H, 65.1% vs. 67.4%, P = 1). There was no significant difference in the nodule detection sensitivity of VCAR between the DLIR-LDCT and DLIR-ULDCT images (72.1% and 67.4%, P = .50).

There was no significant difference of nodule detection sensitivity of VCAR and LuCAS (VCAR vs. LuCAS, 72.1% vs. 86.0%, P = .18) (S3 Table). However, LuCAS showed better performance than VCAR in detection of subsolid nodules (VCAR vs. LuCAS, 14.3% vs. 100%, P = .03) (S3 Table).

Low-dose CT with ASIR-V$_{50}$ Low-dose CT with DLIR-H

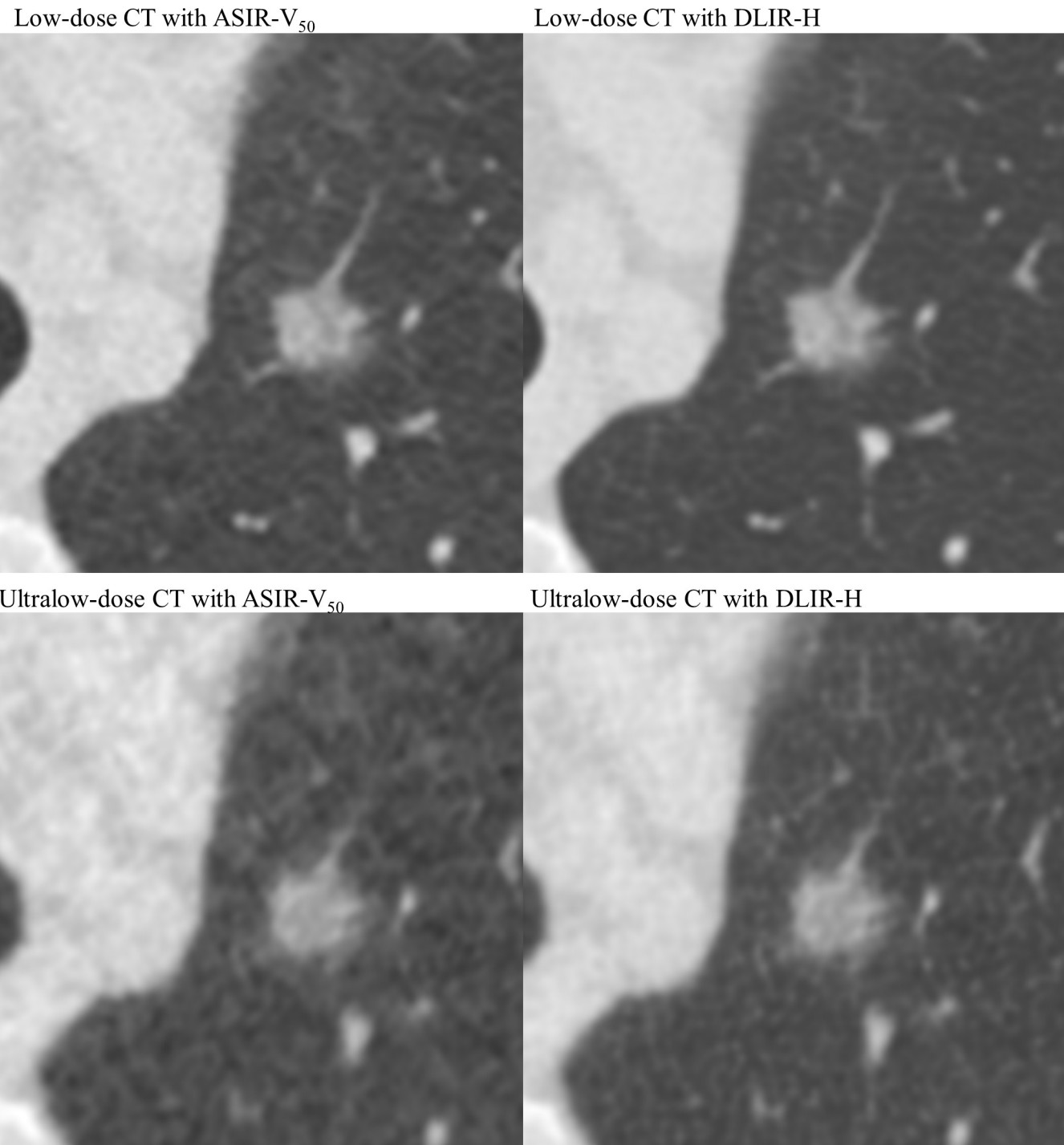

Ultralow-dose CT with ASIR-V$_{50}$ Ultralow-dose CT with DLIR-H

**Fig 3. Representative images of a subsolid nodule in Low-dose and Ultralow-dose CT scans with iterative and deep-learning reconstruction algorithms.**
A 72-year-old female patient with a 1.2-cm subsolid nodule in the left upper lobe underwent low-dose and ultralow-dose chest CT scans. Each CT scan was reconstructed with adaptive statistical iterative reconstruction-V 50% (ASIR-V$_{50}$) and deep-learning image reconstruction (DLIR). DLIR showed superior lesion conspicuity and small vessel visibility compared to the ASIR-V$_{50}$ reconstruction algorithm in both low-dose and ultralow-dose chest CT scans.

Low-dose CT with ASIR-V$_{50}$　　　　　Low-dose CT with DLIR-H

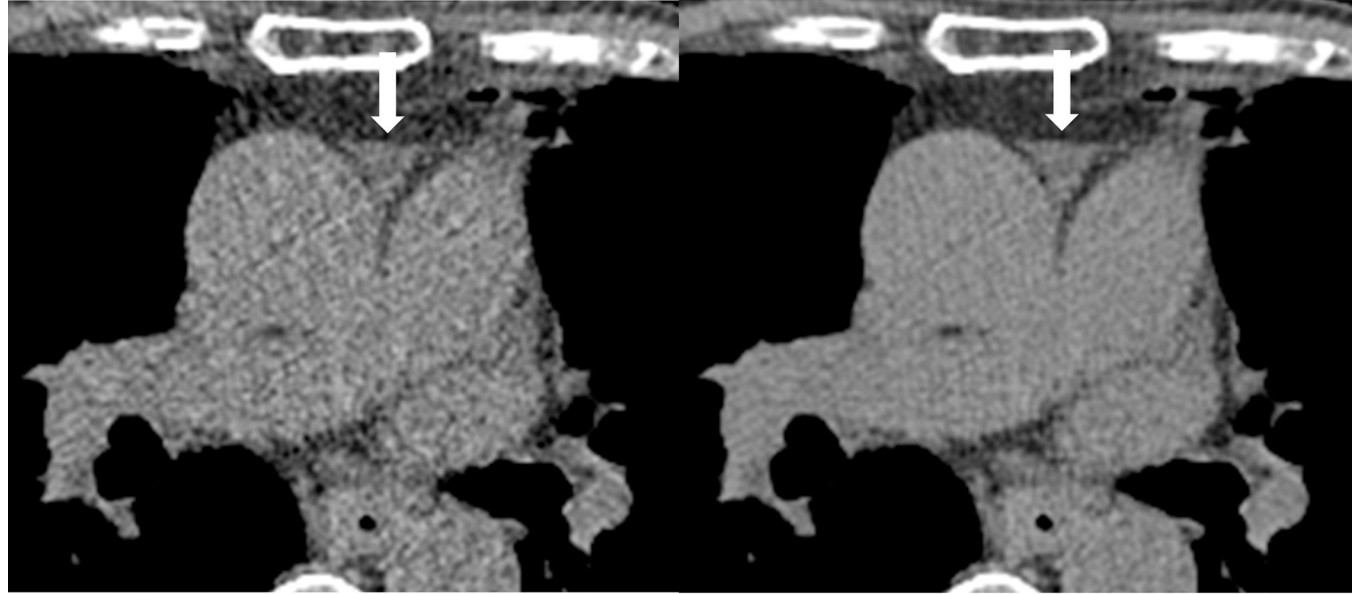

Ultralow-dose CT with ASIR-V$_{50}$　　　　　Ultralow-dose CT with DLIR-

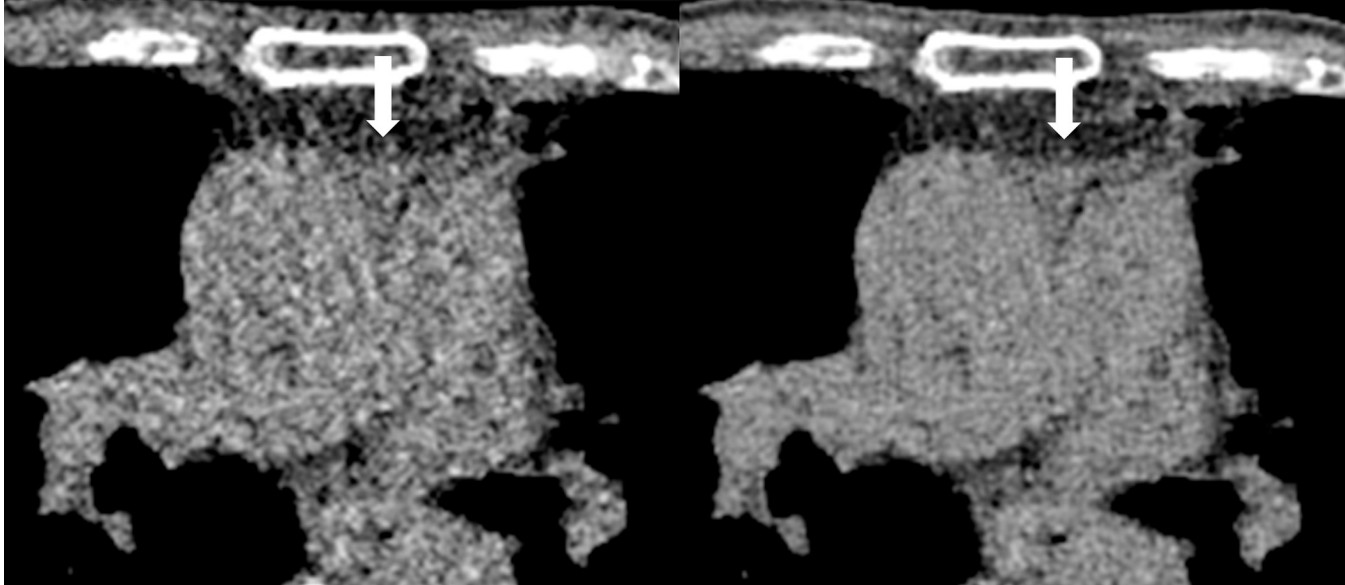

**Fig 4. Representative images of mediastinal structures in Low-dose and Ultralow-dose CT scans with iterative and deep-learning reconstruction algorithm.** A 80-year-old male patient with small amount of pericardial effusion at anterior superior aortic recess (arrow). The boundaries of anterior superior aortic recess is well defined in the DLIR compared to ASIR-V$_{50}$ reconstructed images in both low-dose and ultralow-dose chest CT scans. The boundaries of anterior superior aortic recess is indistinguishable with the aorta and main pulmonary artery in ULDCT scan with ASIR-V$_{50}$.

In total, 26 lung nodules (solid, 11; subsolid, 1; calcified, 14) were detected and volume measured by VCAR in all ASIR-V$_{50}$-LDCT, DLIR-LDCT, and DLIR-ULDCT reconstruction images. When applying the Lung-RADS 1.1 categorization of noncalcified nodules based on nodule volume, there were 10 category 2 nodules, one category 3 nodule, and one category 4B nodule in the ASIR-V$_{50}$-LDCT images (Fig 1). The Lung-RADS 1.1 categories of the 12 nodules were exactly the same among ASIR-V$_{50}$-LDCT, DLIR-LDCT, and DLIR-ULDCT.

**Table 3. Nodule detection sensitivity of radiologists.**

| | DLIR-LDCT | DLIR-ULDCT |
|---|---|---|
| **Per-patient sensitivity** | | |
| Reader 1 | 69.2% (9/13) | 61.5% (8/13) |
| Reader 2 | 100% (13/13) | 100% (13/13) |
| **Per- patient specificity** | | |
| Reader 1 | 96.3% (26/27) | 96.3% (26/27) |
| Reader 2 | 55.6% (15/27) | 66.7% (18/27) |
| **Per-nodule sensitivity** | | |
| Reader 1 | 70.6% (12/17) | 64.7% (11/17) |
| Reader 2 | 88.2% (15/17) | 82.4% (14/17) |
| **Per-nodule sensitivity for subsolid nodules** | | |
| Reader 1 | 100% (7/7) | 85.7% (6/7) |
| Reader 2 | 71.4% (5/7) | 71.4% (5/7) |

Data in parentheses are the number of patients or nodules.

LDCT = low-dose chest computed tomography, ULDCT = ultralow-dose chest computed tomography,

DLIR-H = deep-learning image reconstruction-high level.

## Nodule volume and diameter difference by VCAR

Bland-Altman plots between the measurements of the 26 nodules which were detected and volume measured in all ASIR-V50-LDCT, DLIR-LDCT, and DLIR-ULDCT reconstruction images are shown in Figs 5 and S1. The mean measurement differences and 95% limits of agreement of nodule volumetry were 7.3 mm$^3$ (-63.9, 78.5 mm$^3$) between DLIR- and ASIR-V$_{50}$-LDCT, 3.9 mm$^3$ (-33.9, 41.6 mm$^3$) between DLIR-ULDCT and ASIR-V$_{50}$-LDCT, and -3.5 mm$^3$ (-85.6, 78.7 mm$^3$) between DLIR-LDCT and DLIR-ULDCT (Fig 5). The distribution of nodule volume and x-axis diameter differences are shown in Figs 6 and S2.

17 nodules were detected and volume measured in both Lung VCAR and LuCAS. The mean measurement difference and 95% limits of agreement of nodule volume was 8.0 mm$^3$ (-146.6, 162.5 mm$^3$). Bland-Altman plots between the volume measurements of the nodules is in S3 Fig.

A volume difference greater than 25% occurred in 11.5% of comparisons between DLIR-LDCT and DLIR-ULDCT scans and in 7.7% of comparisons between DLIR- and ASIR-V$_{50}$-LDCT scans; this proportion was not significantly different (P = .65) (number of nodules with measurement difference of volume within 25% between ASIR-V$_{50}$-LDCT and

**Table 4. Nodule detection sensitivity of Lung VCAR.**

| Nodules | LDCT | | ULDCT | |
|---|---|---|---|---|
| | ASIR-V$_{50}$ | DLIR-H | ASIR-V$_{50}$ | DLIR-H |
| Total | 72.1% (31/43) | 72.1% (31/43) | 65.1% (28/43) | 67.4% (29/43) |
| Solid | 77.8% (14/18) | 77.8% (14/18) | 72.2% (13/18) | 72.2% (13/18) |
| Subsolid | 14.3% (1/7) | 14.3% (1/7) | 0% (0/7) | 14.3% (1/7) |
| Calcified | 88.9% (16/18) | 88.9% (16/18) | 83.3% (15/18) | 83.3% (15/18) |

Data in parentheses in number of nodules.

VCAR = volume computerized assisted reporting, LDCT = low-dose chest computed tomography, ULDCT = ultralow-dose chest computed tomography,

ASIR-V = adaptive statistical iterative reconstruction-V, DLIR-H = deep-learning image reconstruction-high level.

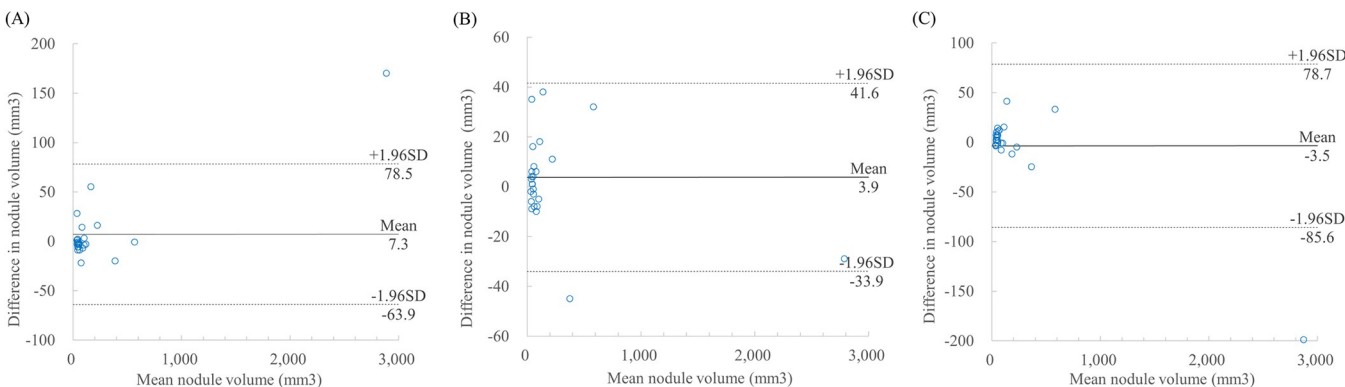

**Fig 5. Bland-Altman plots of nodule volume measurement differences.** Compared to low-dose chest CT with adaptive statistical iterative reconstruction-V 50% (ASIR-$V_{50}$), low-dose chest CT with deep-learning image reconstruction (DLIR) and ultralow-dose chest CT with DLIR showed a mean nodule volume measurement difference of 7.3 mm$^3$ (-63.9, 78.5) (A) and 3.9 mm$^3$ (-33.9, 41.6), respectively (B). In terms of deep learning image reconstruction, low-dose chest CT and ultralow-dose chest CT showed mean nodule volume measurement difference of -3.5 mm$^3$ (-85.6, 78.7) (C). LDCT = low-dose chest computed tomography, ULDCT = ultralow-dose chest computed tomography, ASIR-$V_{50}$ = adaptive statistical iterative reconstruction-V 50%, DLIR = deep-learning image reconstruction, VCAR = volume computerized assisted reporting.

DLIR-LDCT, 24/26; between ASIR-$V_{50}$-LDCT and DLIR-ULDCT, 22/26; and between DLIR-LDCT and DLIR-ULDCT reconstruction, 23/26; P = .39-.68).

Furthermore, 76.9–80.7% of the nodules showed diameter differences equal to or less than 1.5 mm, and the proportions were not significantly different (number of nodules with a measurement difference of the diameter in the x-axis within 1.5 mm between ASIR-$V_{50}$-LDCT and DLIR-LDCT, 21/26; between ASIR-$V_{50}$-LDCT and DLIR-ULDCT, 21/26; and between DLIR-LDCT and DLIR-ULDCT reconstruction, 20/26; P = .74–1).

## Discussion

DLIR provided better SNR and subjective image scores in both LDCT and ULDCT, than ASIR-$V_{50}$. In the reader study, nodule detection sensitivities did not significantly differ between DLIR-LDCT and DLIR-ULDCT (70.6%-88.2% versus 64.7%-82.4%, P = 1). The Lung-RADS 1.1 categories showed substantial agreement between DLIR-LDCT and DLIR-ULDCT for each reader ($\kappa$ = 0.73–0.76). In terms of automated nodule detection by VCAR, there was no significant difference in nodule detection sensitivities between the DLIR-LDCT and DLIR-ULDCT images (72.1% and 67.4%, P = .50). The frequencies of volume change >25% (11.5%, 3/26) or diameter change >1.5mm (23.1%, 6/26) between DLIR-LDCT and DLIR-ULDCT were similar to those (7.7%, 2/26 and 19.2%, 5/26) that occurred between the two different reconstruction algorithms of LDCT (P = .65 and P = .74). One nodule showed a 112% nodule volume difference between ASiR-$V_{50}$-LDCT and DLIR-LDCT and a 140% nodule volume difference between ASiR-$V_{50}$-LDCT and DLIR-ULDCT, but a 13% nodule volume difference between DLIR-LDCT and DLIR-ULDCT. The nodule was the smallest among the 26 nodules. The Lung VCAR measured volume of the nodule was 25 mm$^3$ in ASIR-$V_{50}$-LD CT, 53 mm$^3$ in DLIR-LDCT, and 60 mm$^3$ in DLIR-ULDCT. A larger measurement error in small-sized nodules by CAD was found in previous studies [16,22,23]. However, the Lung-RADS 1.1 category of this small-sized solid nodule was 2, and despite the large percentage volume difference among ASIR-$V_{50}$-LD CT, DLIR-LDCT, DLIR-ULDCT, the Lung-RADS 1.1 category did not change. Our results indicate that DLIR-ULDCT provided a comparable nodule detection sensitivity and Lung-RADS categorization to DLIR-LDCT in both the reader study and VCAR analysis.

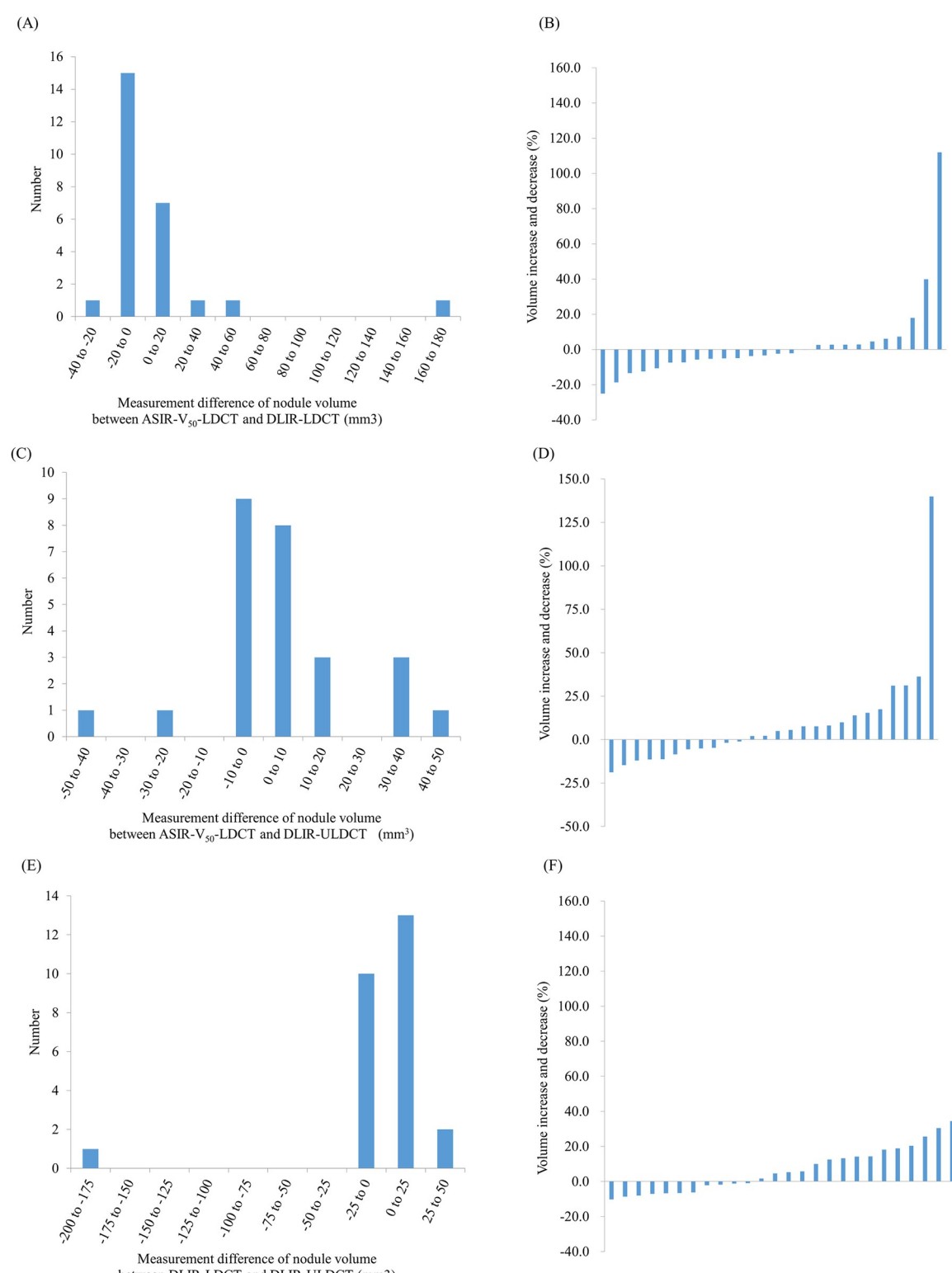

**Fig 6. Distribution of measurement differences of the nodule volume.** Distribution of measurement differences between low-dose chest CT with adaptive statistical iterative reconstruction-V 50% (ASIR-V$_{50}$) and low-dose chest CT with deep-learning image reconstruction (DLIR) (A, B), low-dose chest CT with ASIR-V$_{50}$ and ultralow-dose chest CT with DLIR (C, D), and low-dose chest CT and ultralow-dose chest CT with DLIR (E, F). The majority (84.6–92.3%) of the nodules showed measurement differences of volume within 25%. LDCT = low-dose chest computed tomography, ULDCT = ultralow-dose chest computed tomography, ASIR-V50 = adaptive statistical iterative reconstruction-V 50%, DLIR = deep-learning image reconstruction.

The Lung-RADS 1.1 categories based on the volume of 12 nodules detected by VCAR were consistent among ASIR-V$_{50}$-LDCT, DLIR-LDCT, and DLIR-ULDCT. In the observer performance study, of the 17 nodules with Lung-RADS 1.1 category 3 or 4, three and two nodules were down-categorized or not detected in DLIR-ULDCT compared to DLIR-LDCT by readers 1 and 2, respectively. Two nodules were up-categorized in DLIR-ULDCT compared to DLIR-LDCT by reader 2. Therefore, the nodule management was unchanged in 76–82% of cases. The inter- and intra-observer variability of Lung-RADS categorization is a challenge due to the closeness of categories that are subdivided into units of millimeters or millimeters cubed. In real clinical practice, measurement variability is inevitable. In a previous study, inter-observer disagreement of Lung-RADS categorization was seen in 29% of cases, and 8% resulted in different patient management [24]. Considering this variability, the agreement of Lung-RADS categories in our study is acceptable.

According to the diagnostic reference level (DRL) guidelines published by the Korea Centers for Disease Control and Prevention in the Ministry of Health and Welfare in 2019, the DRLs for chest radiograph posteroanterior and lateral views were 0.40 mGy and 1.26 mGy; considering the tissue-weighting factor of the lung (0.12), the effective doses are 0.05 mSv and 0.15 mSv, respectively [21]. The effective dose of our ULDCT scans was 0.07±0.01mSv, slightly higher than that of posteroanterior chest radiographs and slightly lower than that of chest lateral views. The radiation dose of ULDCT was one-eighth of the LDCT dose. Therefore, ULDCT with DLIR can be considered in situations requiring frequent follow-up of lung lesions.

There are previous studies of a vendor-specific and vendor-agnostic DLIR to create high quality images of ULDCT [25,26]. In a study by Goto et al., ULDCT with vendor-specific DLIR (AiCE, Canon Medical Systems) showed better nodule CNR and comparable subjective image quality assessment (noise in air, noise in soft-tissue, streak artifact, texture fineness, and overall quality) compared to LDCT with iterative reconstruction in the phantom study [25]. In a subjective evaluation, ULDCT with DLIR of 14 nonobese patients achieved a significantly higher preference by radiologists compared to ULDCT with iterative reconstruction [25]. Hata et al. proved a vendor-agnostic DLIR for noise reduction of ULDCT showed better subjective image quality compared to the ULDCT with iterative reconstruction [26]. Also, nodule detection rate of radiologists improved in ULDCT with DLIR compared to ULDCT with iterative reconstruction [26]. These previous studies and ours both align with the same result that ULDCT with DLIR increases objective and subjective image quality compared to iterative reconstruction algorithms and provides image quality comparable to that of LDCT.

There have been concerns regarding the increased radiation exposure from medical imaging of pediatric populations [27]. Efforts to reduce the radiation dose in pediatric CT imaging continued steadily and dose reduction strategies in CT imaging is expected to dramatically reduce the adverse effects of radiation exposure such as radiation-induced cancers [27,28]. One of the proven dose reduction strategies is DLIR which can minimize image noise while preserving the adequate resolution of the CT scans [28,29]. In a study by Yoon et al., in pediatric non-contrast and contrast-enhanced chest CT (CTDIvol, 1.3 ± 0.5 mGy; DLP 49.0 ± 26.3 mGy × cm; effective dose, 2.2 ± 3.2 mSv), DLIR showed significant noise reduction with increased SNR and CNR and showed better subjective overall image quality and noise compared to iterative reconstruction [29]. This result is comparable to our results and application of DLIR is expected to substantially reduce the radiation dose while maintaining the appropriate image quality for diagnosis in pediatric chest CT scans.

Our study has a few limitations. First, we prospectively included a limited number of patients with respiratory disease in this study during the early pandemic. Second, for automated nodule detection and measurement evaluation, we inevitably excluded patients with

more than 20 lung nodules because setting the reference was not practically possible. Although the automated nodule detection tool shows a high nodule detection rate, in CT scans with diffuse nodular disease, a reduced specificity would be inevitable [30]. Therefore, the results of our study may have limited applicability to patients with diffuse nodular diseases. Regarding the volume measurements, we did not allow readers to modify the VCAR's nodule segmentation. If human readers could further edit the VCAR's measurements, it would help maintain more consistent volumetric measurements between CT scans. Third, in the reliability evaluation of Lung VCAR, ground-glass nodule detection sensitivity of the VCAR was inferior to LuCAS. However the two software had no significant statistical difference in total lung nodule detection sensitivity. The nodule volume measurement difference between the two software was acceptable (mean ± standard deviation, $8.0 \pm 78.9$ mm$^3$). Therefore, we determined that using Lung VCAR for CT analysis is reasonable. Lastly, we included non-obese patients to consistently maintain the CT parameters and the CTDIvol of 0.1 mGy for a ULDCT scan as much as possible. Obese patients occupied 28% of the NLST screening population [31] and required a three to five times higher CT exposure than our protocol for a ULDCT scan [32,33]. Up-to-date studies evaluating a ULDCT with DLIR included non-obese patients [13,14] and future study is warranted for the applicability of ULDCT with DLIR for obese patients.

In conclusion, our study demonstrated that DLIR showed better subjective and objective image quality than iterative reconstruction. In terms of the nodule detection rate and Lung-RADS 1.1 categories of the nodules, DLIR-ULDCT was comparable to DLIR-LDCT in both the reader study and VCAR analysis, while the radiation dose was similar to that of chest radiographs. ULDCT with DLIR may be a reasonable option for monitoring pulmonary nodules using Lung-RADS or the VCAR system in patients who require repeated CT scans, with an as low as reasonably achievable chest X-ray dose.

## Supporting information

**S1 Fig. Bland-Altman plots of the X-axis diameter difference of lung nodules.** Compared to low-dose chest CT with adaptive statistical iterative reconstruction-V 50% (ASIR-V$_{50}$), low-dose chest CT with deep-learning image reconstruction (DLIR) and ultralow-dose chest CT with DLIR showed mean x-axis diameter differences of 0.4 mm (-4.0, 4.8) (a) and -0.1 mm (-2.2, 2.0), respectively (b). In terms of deep learning image reconstruction, low-dose chest CT and ultralow-dose chest CT showed a mean x-axis diameter difference of -0.5 mm (-5.0, 4.0) (c). LDCT = low-dose chest computed tomography, ULDCT = ultralow-dose chest computed tomography, ASIR-V$_{50}$ = adaptive statistical iterative reconstruction-V 50%, DLIR = deep-learning image reconstruction, VCAR = volume computerized assisted reporting.
(TIF)

**S2 Fig. Distribution of measurement differences of the X-axis diameter of lung nodules.** Distribution of measurement differences between low-dose chest CT with adaptive statistical iterative reconstruction-V 50% (ASIR-V$_{50}$) and low-dose chest CT with deep-learning image reconstruction (DLIR) (a,b), low-dose chest CT with ASIR-V$_{50}$ and ultralow-dose chest CT with DLIR (c, d), and low-dose chest CT and ultralow-dose chest CT with DLIR-H (e, f). In total, 76.9–80.7% of the nodules showed a measurement difference of the diameter equal to or less than 1.5 mm. LDCT = low-dose chest computed tomography, ULDCT = ultralow-dose chest computed tomography, ASIR-V$_{50}$ = adaptive statistical iterative reconstruction-V 50%, DLIR = deep-learning image reconstruction.
(TIF)

**S3 Fig. Bland-Altman plots of the volume difference of lung nodules in ASIR-V$_{50}$-LDCT between lung VCAR and LuCAS.** ASIR-V$_{50}$ = adaptive statistical iterative reconstruction-V 50%, LDCT = low-dose chest computed tomography, VCAR = volume computerized assisted reporting.
(TIF)

**S1 Table. Scoring system of subjective image quality.**
(DOCX)

**S2 Table. Results of the observer performance study for the detection of nodules with Lung-RADS 1.1 Categories 3 and 4.**
(DOCX)

**S3 Table. Nodule detection sensitivity of Lung VCAR and LuCAS applied on low-dose chest ct reconstructed with ASIR-V$_{50}$.**
(DOCX)

## Author Contributions

**Conceptualization:** Soon Ho Yoon.

**Data curation:** Seung-Jin Yoo, Young Sik Park, Soon Ho Yoon.

**Formal analysis:** Seung-Jin Yoo, Soon Ho Yoon.

**Funding acquisition:** Soon Ho Yoon.

**Investigation:** Seung-Jin Yoo, Soon Ho Yoon.

**Methodology:** Seung-Jin Yoo, Soon Ho Yoon.

**Project administration:** Soon Ho Yoon.

**Resources:** Young Sik Park, Soon Ho Yoon.

**Software:** Seung-Jin Yoo, Soon Ho Yoon.

**Supervision:** Soon Ho Yoon.

**Validation:** Seung-Jin Yoo, Hyewon Choi, Da Som Kim, Soon Ho Yoon.

**Visualization:** Seung-Jin Yoo, Soon Ho Yoon.

**Writing – original draft:** Seung-Jin Yoo, Soon Ho Yoon.

**Writing – review & editing:** Seung-Jin Yoo, Young Sik Park, Hyewon Choi, Da Som Kim, Jin Mo Goo, Soon Ho Yoon.

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
