## [Decision Letter · Decision Letter 0]

7 Jul 2023

PONE-D-23-05687Prospective Evaluation of Deep Learning Image Reconstruction for Lung-RADS and Automatic Nodule Volumetry on Ultralow-Dose Chest CTPLOS ONE

Dear Dr. Yoon:

Thank you for submitting your manuscript to PLOS ONE. After careful consideration, we feel that it has merit but does not fully meet PLOS ONE’s publication criteria as it currently stands. Therefore, we invite you to submit a revised version of the manuscript that addresses the points raised during the review process.

Revisions were proposed by both reviewers.  Please also note that improvements in the figures were recommended for the next revision.

We look forward to receiving your revised manuscript.

Kind regards,

Gayle E. Woloschak, PhD

Section Editor

PLOS ONE

Journal Requirements:

“This work was supported by GE Healthcare. The funder had no role in study design, data collection and analysis, decision to publish, or preparation of the manuscript.”

“Soon Ho Yoon works in the MEDICALIP as a chief medical officer.

The other author do not have any conflicts of interest to declare for this article.”

We note that one or more of the authors are employed by a commercial company: MEDICALIP

A.            Please provide an amended Funding Statement declaring this commercial affiliation, as well as a statement regarding the Role of Funders in your study. If the funding organization did not play a role in the study design, data collection and analysis, decision to publish, or preparation of the manuscript and only provided financial support in the form of authors' salaries and/or research materials, please review your statements relating to the author contributions, and ensure you have specifically and accurately indicated the role(s) that these authors had in your study. You can update author roles in the Author Contributions section of the online submission form.

B. Please also provide an updated Competing Interests Statement declaring this commercial affiliation along with any other relevant declarations relating to employment, consultancy, patents, products in development, or marketed products, etc. 

Additional Editor Comments (if provided):

One reviewer suggested major revisions, the other suggested minor changes. Please work to address all of these concerns in a revision. Also note that there were concerns about the quality of figures which need to be improved.

Reviewers' comments:

Reviewer's Responses to Questions

**Comments to the Author**

1. Is the manuscript technically sound, and do the data support the conclusions?

Reviewer #1: Partly

Reviewer #2: Yes

2. Has the statistical analysis been performed appropriately and rigorously? 

Reviewer #1: No

Reviewer #2: Yes

3. Have the authors made all data underlying the findings in their manuscript fully available?

Reviewer #1: Yes

Reviewer #2: No

4. Is the manuscript presented in an intelligible fashion and written in standard English?

Reviewer #1: Yes

Reviewer #2: Yes

5. Review Comments to the Author

Reviewer #1: This study evaluated whether lung nodule detection, Lung-RADS classification, and volumetric nodule

assessments were feasible with ULD chest CT scans with DLIR. Study included 40 patient cases reviewed by 2 observers with a commercialized assisted tool.

Authors concluded DLIR enabled comparable Lung-RADS and volumetric nodule assessments on

ULDCT images to LDCT images.

1. Study included assessment with a commercialized assisted tool (VCAR) in addition to observers' evaluation. Authors did not state the rationale of including the assessment using a software tool, and the reason why VCAR was introduced in this study is not clear.

If authors intended to use VCAR as a detection and measurement device, and compared nodule assessment performance by VCAR for LDCT and ULDCT scans with IR and DLIR, then the performance reliability of VCAR in terms of nodule detection and volume measurement need to be first assessed before drawing any conclusion based on measurements using that particular device. And yet, I could not find any description on that.

Authors should make it clear the role of VCAR assessment in this study and the way how its reliability was assessed.

2. In Fig. 5 (A) and (B), VCAR measurements already differ btw ASiR-LDCT and DLIR-LDCT quite a lot, which indicate VCAR's performance is not sufficiently reliable to distinguish between LDCT and ULDCT. Authors need to present the reliability range of VCAR's performance in order to justify its use in this study purpose.

3. In Fig. 5 (B) and (C), volume measurement difference of case no. 17 exceeded 100%, which may result in a different patient management and represent a potential risk. And yet, authors did not mention that risk. Study results should be discussed in both positive and negative aspects.

4. M&M section states a total of 43 nodules were assessed with VCAR, but the number is 27 in Fig. 5. Needs to be clarified.

5. Authors stated that there was no significant difference in the nodule detection sensitivity between the DLIR-LDCT and DLIR-ULDCT images. I assume authors used a chi-squire test for calculating the statistical significance, which is not clear to the reviewer. Please explain the way in detail.

6. For nodule detection study with observers, how were the gold standard nodules established ?

Reviewer #2: LDCT reduces lung cancer mortality, but it needs annual/short-term follow-ups which deliver radiation exposures.a ULDCT is 6-10 times less dose than LDCT, and DLIR were shown to provide better image quality. This work attempts to evaluate classification and evaluation of the images, and the authors reached this goal using DLIR for hospital measurements. Given that it is informative on using DLIR for Ultra low dose chest CT, I think readers will benefit from this manuscript for their own practices. Before recommending for publication, I only have a few minor concerns on this manuscript.

Introduction: "better image quality ... than the preexisting iterative reconstruction method (13-15)" suggests one single method but with multiple citations. Do the authors mean multiple methods?

Study population: is the period of Feb-Jul 2020 chosen on purpose, because it overlaps with the onset of COVID-19 pandemic?

Objective and subjective image quality assessment: the authors have four thoracic radiologists, with two for each task. Is the number too small for the assessment, if their assessment results diverge significantly?

Statistical analysis:

(1) For the Bonferroni correction, why is 6 used?

(2) For ICC thresholds are different across different studies, why did the authors choose 0.40 and 0.75? If this is a field-specific choice, can the authors provide citation to the choice?

(3) Does Fleiss' kappa offer a different result, if offered?

Table 1: what are the two values in the brackets of 21 (53) and 19 (47)?

Table 2: * - could the mention of Bonferroni here be more clear for the adopted significance level?

Nodule detectability and Lung-RADS categorization by VCAR: given that there is no significant difference between the two sets of images, would the authors like to mention that DLIR-ULDCT should be preferred given its ultra low dose?

Discussion: nice summary of the findings and the limitations.

Figure 3: can the authors highlight certain features that were not visible with ASIR-V50 but evident with DLIR?

6. PLOS authors have the option to publish the peer review history of their article (what does this mean?). If published, this will include your full peer review and any attached files.

Reviewer #1: No

Reviewer #2: No

---

## [Author Response · Author response to Decision Letter 0]

22 Oct 2023

Reviewer #1: This study evaluated whether lung nodule detection, Lung-RADS classification, and volumetric nodule assessments were feasible with ULD chest CT scans with DLIR. Study included 40 patient cases reviewed by 2 observers with a commercialized assisted tool.

Authors concluded DLIR enabled comparable Lung-RADS and volumetric nodule assessments on ULDCT images to LDCT images.

1. Study included assessment with a commercialized assisted tool (VCAR) in addition to observers' evaluation. Authors did not state the rationale of including the assessment using a software tool, and the reason why VCAR was introduced in this study is not clear.

If authors intended to use VCAR as a detection and measurement device, and compared nodule assessment performance by VCAR for LDCT and ULDCT scans with IR and DLIR, then the performance reliability of VCAR in terms of nodule detection and volume measurement need to be first assessed before drawing any conclusion based on measurements using that particular device. And yet, I could not find any description on that.

Authors should make it clear the role of VCAR assessment in this study and the way how its reliability was assessed.

Response: Thank you for the keen comment.

Although there is a variety of software for automated detection and measurement of lung nodules in chest CT scans, the reason why we choose to use Lung VCAR is as follows: 

First, among the various software for nodule detection and measurement for Chest CT scans, we thought using FDA-approved VCAR software from the same vendor (GE Healthcare) with our CT scanner could ensure consistency to this study. The purpose of our study is to compare the quality of DLIR compared to ASIR-V50 reconstruction and ULDCT to LDCT rather than evaluate the CAD software’s nodule detection ability and measurement accuracy. 

Second, a previous study done by Lee et al. (Medicine 2020;99:23(e20543)) showed acceptable performance of Lung VCAR in nodule volume measurement with absolute percentage volume error around 10% in low-dose and standard-dose chest CT scans reconstructed with FBP and ASIR-V. Liang et al. (AJR 2017; 209:304–308) compared the volume of stable nodules in chest CT scans using two commercial software, Lung VCAR and SyngoVia. Lung VCAR showed a mean volume variation of 0.7% ± 18.6% (range, −39.3% to 83.3%) which was better than SyngoVia. These previous studies supported that Lung VCAR was reliable and became for our purpose.

Third, we initially assessed the performance of nodule detection by Lung VCAR for LDCT and ULDCT scans reconstructed with FBP, ASIR-V50, ASIR-V100 and DLIR and the nodule detection sensitivity showed consistent results, which was sufficient for analysis. The nodule detection sensitivities are listed in the table below.

 LDCT ULDCT

Nodules FBP ASIR-V 50 ASIR-V 100 DLIR-H FBP ASIR-V 50 ASIR-V 100 DLIR-H

Total 0.70 (30/43) 0.72 (31/43) 0.74 (32/43) 0.72 (31/43) 0.63 (27/43) 0.65 (28/43) 0.65 (28/43) 0.67 (29/43)

Solid 0.78 (14/18) 0.78 (14/18) 0.78 (14/18) 0.78 (14/18) 0.67 (12/18) 0.72 (13/18) 0.72 (13/18) 0.72 (13/18)

Subsolid 0.00 (0/7) 0.14 (1/7) 0.14 (1/7) 0.14 (1/7) 0.00 (0/7) 0.00 (0/7) 0.00 (0/7) 0.14 (1/7)

Calcified 0.89 (16/18) 0.89 (16/18) 0.94 (17/18) 0.89 (16/18) 0.83 (15/18) 0.83 (15/18) 0.83 (15/18) 0.83 (15/18)

In response to your comment, we further thought about solutions to evaluate Lung VCAR’s reliability. Identifying a gold standard for nodule detection sensitivity and measurement was challenging since every software has limitations. Even in experienced radiologists, inter and intra-observer variability regarding nodule detection and measurement is inevitable. 

We have compared VCAR with another commercially available software called LuCAS (Monitor Corporation, Seoul, Korea). LuCAS is a deep learning-based computer-aided diagnosis (DL-CAD) for nodule detection and segmentation in chest CT scans. The nodule detection sensitivity in LDCT with ASIR-V50 reconstruction showed no statistically significant difference between the two software (VCAR vs. LuCAS, 72% vs. 86%, p=.113). However, LuCAS performed better than VCAR in detecting subsolid nodules.

 LDCT – ASIR-V50

Nodules VCAR LuCAS

Total 0.72 (31/43) 0.86 (37/43)

Solid 0.78 (14/18) 0.78 (14/18)

Subsolid 0.14 (1/7) 1.00 (7/7)

Calcified 0.89 (16/18) 0.89 (16/18)

The mean ± standard deviation of nodule volume difference between VCAR and LuCAS was 8.0 ± 78.9 mm3. The Bland-Altman plot below shows an acceptable measurement difference between the two software.

2. In Fig. 5 (A) and (B), VCAR measurements already differ btw ASiR-LDCT and DLIR-LDCT quite a lot, which indicate VCAR's performance is not sufficiently reliable to distinguish between LDCT and ULDCT. Authors need to present the reliability range of VCAR's performance in order to justify its use in this study purpose.

Response: Thank you for the comment. The answer to this comment will be presented together in questions 1 and 3.

3. In Fig. 5 (B) and (C), volume measurement difference of case no. 17 exceeded 100%, which may result in a different patient management and represent a potential risk. And yet, authors did not mention that risk. Study results should be discussed in both positive and negative aspects.

Response: Thank you for your keen comment. 

In the waterfall charts of Figure 5 (B) and (D), one nodule’s volume difference exceeds 100% between ASIR-V50-LDCT and DLIR-LDCT and between ASIR-V50-LDCT and DLIR-ULDCT. The nodule was the smallest among the 26 nodules: a 4.5 mm-sized solid nodule. The Lung VCAR measured nodule volume was 25 mm3 in ASIR-V50-LD CT, 53 mm3 in DLIR-LDCT, and 60 mm3 in DLIR-ULDCT. Since the nodule is a small-sized solid nodule, Lung-RADS 1.1 category was 2 and despite the large percentage volume difference among ASIR-V50-LD CT, DLIR-LDCT, DLIR-ULDCT, the Lung-RADS 1.1 category did not change. 

Previous studies also found larger measurement error in small-sized nodules by CAD and manual measurement by radiologists (AJR 2017; 209:304–308. Medicine 2020;99:23(e20543). Eur Radiol 2023;33:5568–5577.). This reaffirms that such inconsistencies aren't solely a limitation of the CAD software but are observed across different measurement techniques, particularly for smaller nodules.

In our study, Lung-RADS 1.1 category of 12 nodules with categories 3 and 4 stayed the same among ASIR-V50-LDCT, DLIR-LDCT, and DLIR-ULDCT. CAD software, including Lung VCAR, has limitations, but if used in the appropriate and specific situations, it would be helpful in clinical practice. 

4. M&M section states a total of 43 nodules were assessed with VCAR, but the number is 27 in Fig. 5. Needs to be clarified.

Response: Thank you for pointing out the discrepancy. To clarify, we assessed the nodule detection sensitivity of VCAR for a total of 43 nodules. Out of theses, 26 nodules were consistently detected and their volume were measured by VCAR in all ASIR-V50-LDCT, DLIR-LDCT, and DLIR-ULDCT reconstruction images. Therefore, the comparisons in Lung-RADS 1.1 categorization and nodule volume difference were focused on these 26 nodules. Bland-Altman plots in Figure 4 and waterfall charts in Figure 5 show the 26 nodules’ volume difference. You are right about the number 27 in Figure 5d - it was an oversight on our part and has now been corrected to “26”. We apologize for any confusion this may have caused and appreciate your keen observation.

5. Authors stated that there was no significant difference in the nodule detection sensitivity between the DLIR-LDCT and DLIR-ULDCT images. I assume authors used a chi-squire test for calculating the statistical significance, which is not clear to the reviewer. Please explain the way in detail.

Response: We used the chi-square test for calculating the nodule detection sensitivity of each radiologist in DLIR-LDCT and DLIR-ULDCT scans. We added the detail in the statistical analysis as follows.

“ The chi-square test was conducted to compare the nodule detection sensitivities of each radiologist in DLIR-LDCT and DLIR-ULDCT.”

6. For nodule detection study with observers, how were the gold standard nodules established ?

Response: The gold standard nodules for nodule detection and Lung-RADS categorization was done by one board-certified radiologist with 7 years of experience in chest CT interpretation. We added the detail in the manuscript as follows.

“ One board-certified thoracic radiologist (***) reviewed 40 DLIR-LDCT scans thoroughly and detected nodules of Lung-RADS 1.1 category 3 or over for gold standard.”

 

Reviewer #2: LDCT reduces lung cancer mortality, but it needs annual/short-term follow-ups which deliver radiation exposures.a ULDCT is 6-10 times less dose than LDCT, and DLIR were shown to provide better image quality. This work attempts to evaluate classification and evaluation of the images, and the authors reached this goal using DLIR for hospital measurements. Given that it is informative on using DLIR for Ultra low dose chest CT, I think readers will benefit from this manuscript for their own practices. Before recommending for publication, I only have a few minor concerns on this manuscript.

1. Introduction: "better image quality ... than the preexisting iterative reconstruction method (13-15)" suggests one single method but with multiple citations. Do the authors mean multiple methods?

Response: We tried to cite three previous studies showing better image quality of deep-learning image reconstruction compared to the iterative reconstruction method. The citations (13-15) represent different studies, each comparing DLIR with a particular iterative reconstruction method. We apologize for any confusion caused by the wording.

2. Study population: is the period of Feb-Jul 2020 chosen on purpose, because it overlaps with the onset of COVID-19 pandemic?

Response: It was a coincidence. We started to plan this study in December 2019. IRB approval and clinical trial registration were completed in January 2020, before COVID-19 became a pandemic. In our country, the first COVID-19 patient occurred on January 1st, 2020. Due to the unexpected pandemic disease, the patient enrollment period took longer than we expected, but there was no significant impact on our study progress.

3. Objective and subjective image quality assessment: the authors have four thoracic radiologists, with two for each task. Is the number too small for the assessment, if their assessment results diverge significantly?

Response: Thank you for your concern. While we acknowledge that having only two radiologists for each task may seem limited, the reviewers were composed of board-certified thoracic radiologists with sufficient CT interpretation experience. Importantly, ICC showed excellent interobserver agreement for subjective assessment of image noise and diagnostic acceptability (0.82 and 0.86, respectively). Given the strong consistency in their evaluations, we believe their assessments provided meaningful and reliable results for this study.

4. Statistical analysis:

(1) For the Bonferroni correction, why is 6 used?

Response: Bonferroni correction compensates for the multiple comparisons we made in our analysis. For our study, we carried out multiple comparisons across 4 types of CT scans (ASIR-V50-LDCT, ASIR-V50-ULDCT, DLIR-LDCT, DLIR-ULDCT scans). The number of unique pairwise comparisons among these four scans is 6 (calculated as 4C2=6) Hence, we adjusted our significance level as follows:

Bonferroni-corrected p-value = original p-value / number of tests performed

0.008 = 0.05 / 6

(2) For ICC thresholds are different across different studies, why did the authors choose 0.40 and 0.75? If this is a field-specific choice, can the authors provide citation to the choice?

Response: Thank you for the comment. We do agree with the reviewer’s opinion that ICC thresholds differ across studies, therefore reference is necessary. We cited the article written by Domenic V. Cicchetti written in 1994 and added the reference no. 18 for the ICC thresholds in the manuscript.

(3) Does Fleiss' kappa offer a different result, if offered?

Response: Thank you for your keen question. Prior to writing this manuscript, we had statistical consultation. ICC was recommended for the interobserver agreement in subjective image quality assessment of the CT scans, and kappa was not recommended. Since the subjective image quality assessment was a scoring task (ordinal variable), kappa, which is for the nominal variable, was not applicable. Cohen’s Kappa was recommended for the Lung-RADS 1.1 categorization agreement between two radiologists, and Fleiss’ Kappa, which is for interobserver agreement for 3 or more raters, was not applicable. 

5. Table 1: what are the two values in the brackets of 21 (53) and 19 (47)?

Response: The data are the number of patients with percentages in parentheses. We indicated in the footnotes as follows.

“ † Data are number of patients with percentage in parentheses.”

6. Table 2: * - could the mention of Bonferroni here be more clear for the adopted significance level?

Response: Thank you for the comment. We amended our footnotes and mentioned Bonferroni as follows.

“ * Bonferroni corrected p value < .008 indicated a statistically significant difference.”

7. Nodule detectability and Lung-RADS categorization by VCAR: given that there is no significant difference between the two sets of images, would the authors like to mention that DLIR-ULDCT should be preferred given its ultra low dose?

Response: We appreciate your observation. There was no significant difference in the nodule detection sensitivity of VCAR between the DLIR-LDCT and DLIR-ULDCT images (72.1% and 67.4%, P=.64). Lung-RADS 1.1 categorization of 12 nodules of category 3, 4 nodules by VCAR were exactly the same. Therefore, DLIR-ULDCT is a scan comparable to DLIR-LDCT. However, in previous studies, it is known that the measurement accuracy of VCAR is poor when the size of the nodule is small, so the accuracy may be low for category 2 nodules. In actual clinical practice, nodule detection CAD for chest CT scans is not popularized, and nodule detection and risk assessment are usually performed by a radiologist. In our observer study, radiologists’ nodule detection sensitivities did not significantly differ between DLIR-LDCT and DLIR-ULDCT (70.6%-88.2% vs. 64.7%-82.4%, P=.57). In terms of Lung-RADS 1.1 categories, the two radiologists did not detect or down-categorized three and two nodules each in DLIR-ULDCT compared to in DLIR-LDCT. Therefore, we advocate DLIR-ULDCT primarily for follow-up scans for monitoring pulmonary nodules rather than initial assessments. We have incorporated this in the conclusion.

8. Figure 3: can the authors highlight certain features that were not visible with ASIR-V50 but evident with DLIR?

Response: DLIR showed superior lesion conspicuity and small vessel visibility in the lung window setting compared to the ASIR-V50 reconstruction algorithm in both low-dose and ultralow-dose chest CT scans. Additionally, in the mediastinal window setting, mediastinal lesions and anatomical structures were more clearly depicted in DLIR compared to ASIR-V50 reconstruction algorithm in both low-dose and ultralow-dose chest CT scans. We have added some figures in Figure 3 to exemplify these points.

---

## [Decision Letter · Decision Letter 1]

8 Nov 2023

PONE-D-23-05687R1Prospective Evaluation of Deep Learning Image Reconstruction for Lung-RADS and Automatic Nodule Volumetry on Ultralow-Dose Chest CTPLOS ONE

Dear Dr. Yoon,

Thank you for submitting your manuscript to PLOS ONE. After careful consideration, we feel that it has merit but does not fully meet PLOS ONE’s publication criteria as it currently stands. Therefore, we invite you to submit a revised version of the manuscript that addresses the points raised during the review process.

Minor recommendations were submitted by one reviewer.  Please address those in a revision.

We look forward to receiving your revised manuscript.

Kind regards,

Gayle E. Woloschak, PhD

Section Editor

PLOS ONE

Journal Requirements:

Additional Editor Comments:

One reviewer had agreed to accept the work, one had suggested minor revisions. Please address the minor revisions in a resubmission.

Reviewers' comments:

Reviewer's Responses to Questions

**Comments to the Author**

1. If the authors have adequately addressed your comments raised in a previous round of review and you feel that this manuscript is now acceptable for publication, you may indicate that here to bypass the “Comments to the Author” section, enter your conflict of interest statement in the “Confidential to Editor” section, and submit your "Accept" recommendation.

Reviewer #2: All comments have been addressed

Reviewer #3: (No Response)

2. Is the manuscript technically sound, and do the data support the conclusions?

Reviewer #2: Yes

Reviewer #3: Yes

3. Has the statistical analysis been performed appropriately and rigorously? 

Reviewer #2: Yes

Reviewer #3: Yes

4. Have the authors made all data underlying the findings in their manuscript fully available?

Reviewer #2: Yes

Reviewer #3: No

5. Is the manuscript presented in an intelligible fashion and written in standard English?

Reviewer #2: Yes

Reviewer #3: (No Response)

6. Review Comments to the Author

Reviewer #2: (No Response)

Reviewer #3: This article is well-written and interesting, and it holds clinical importance, given that the authors have revealed that DLIR is suitable for low-dose lung CT.

I have several suggestions that the authors need to address.

Introduction

1. There are significant controversies surrounding the actual risk of low-dose radiation exposure for adults. For instance, a recent study has indicated that LDCT does not induce DNA damage (source: https://doi.org/10.1148/radiol.2020190389). Whether LDCT entails "non-negligible radiation exposure" remains unclear. The description requires modification.

Methods

2. It is unclear how to determine the sample size (n=40) for this prospective study. Please provide clarification.

3. Please specify the number of patients excluded based on each exclusion criterion.

CT acquisition parameters

4. Please include information about the noise index for both LDCT and ULDCT protocols if the authors utilized automated tube current modulation. This crucial information should be presented in the main text rather than the supporting text.

5. Why did the authors need to treat S2, S3, and S4 texts as supporting text? This important information should be included in the main text, unless there are reasonable constraints such as word count limitations.

6. The McNemar test is recommended for intraindividual comparisons of nodule detection sensitivity, rather than the Chi-square test.

Discussion

7. One of the major limitations in this study is that the authors used a vendor-specific DLIR algorithm, and as a result, the applicability to other DLIR algorithms provided by different vendors is not discussed. Recent investigations employed a different vendor-specific DLR for ULDCT of the lung to improve image quality (doi: 10.1016/j.acra.2022.04.025 and doi: 10.2214/AJR.19.22680.). Please provide a concise discussion of whether your results align with these prior investigations to offer readers the wide applicability of study findings.

8. The study findings may be applicable for pediatric CT, where radiation exposure is of concern　in comparison to adults who require lung cancer screening. Discussing the applicability to low-dose pediatric CT by citing the relevant articles (such as doi: 10.1148/rg.2021210105. and doi: 10.1186/s12880-021-00677-2.) would further emphasize the value of the authors' findings.

7. PLOS authors have the option to publish the peer review history of their article (what does this mean?). If published, this will include your full peer review and any attached files.

Reviewer #2: No

Reviewer #3: No

---

## [Author Response · Author response to Decision Letter 1]

29 Dec 2023

1. There are significant controversies surrounding the actual risk of low-dose radiation exposure for adults. For instance, a recent study has indicated that LDCT does not induce DNA damage (source: https://doi.org/10.1148/radiol.2020190389). Whether LDCT entails "non-negligible radiation exposure" remains unclear. The description requires modification.

Response: Thank you for your comment. The study by Sakane et al. proved that a single low-dose CT with a median effective dose of 1.5mSv does not damage human DNA in peripheral blood lymphocytes. While these findings are innovative and potentially impactful, they do not encompass patient groups who undergo annual or more frequent low-dose CT scans. 

In clinical practice, repeated low-dose CT scans are common due to various conditions, making difficult to ignore the effects of cumulative radiation exposure in these cases. Therefore, clinicians and radiologists continue to make efforts to minimize radiation exposure in patients.

We amended the description as below:

A recent study by Sakane et al. reported no damage of human DNA from a single low-dose CT scan [6]. However, lung cancer CT screening and pulmonary disease evaluations may often involve the frequent use of follow-up CT scans [7], and the safety of the cumulative radiation exposure remains underexplored.

2. It is unclear how to determine the sample size (n=40) for this prospective study. Please provide clarification.

Response: We assumed that deep-learning image reconstruction in ultra low-dose CT scan group will obtain 90% of satisfactory subjective overall quality and iterative reconstruction in ultra low-dose CT scan group will obtain 65% of satisfactory subjective overall quality. Applying the alpha error =0.05, and beta error =0.20 (power=0.80), the calculated total sample size was 40.

3. Please specify the number of patients excluded based on each exclusion criterion.

Response: Since this study was a prospective cross-sectional study, the patient enrollment was done by asking patients who met the inclusion criteria and did not meet the exclusion criteria their willingness to participate in this study and obtaining their consent at a pulmonology outpatient clinic center about. Therefore, the total number of the all potential participants who met the inclusion criteria or met the exclusion criteria was not separately recorded.

4. Please include information about the noise index for both LDCT and ULDCT protocols if the authors utilized automated tube current modulation. This crucial information should be presented in the main text rather than the supporting text.

Response: Thank you for the comment. The noise index for LDCT was 28 and for ULDCT was 33. We relocated the S1 text in the main manuscript and added the noise index as follows:

Consecutive full-inspiratory thoracic LDCT scans at 120 kVp and ULDCT at 100 kVp were acquired in all patients using a single CT machine (Revolution CT; GE Healthcare, Waukesha, WI, USA) and the scanning parameters were as follows: tube voltage, 120 kVp with a mAs of 25 for LDCT, 100 kVp with a mAs of 5 for ULDCT; automatic tube current modulation; gantry rotation time, 280 ms; detector configuration, 128 × 0.625 mm; beam pitch, 1.53; matrix, 512x512; reconstruction increment and section thickness, 1.25 mm; noise index for LDCT, 28 and noise index for ULDCT, 33.

5. Why did the authors need to treat S2, S3, and S4 texts as supporting text? This important information should be included in the main text, unless there are reasonable constraints such as word count limitations.

Response: Thank you for the comment. As you suggested we relocated the S2, S3, S4 text within the main manuscript.

6. The McNemar test is recommended for intraindividual comparisons of nodule detection sensitivity, rather than the Chi-square test.

Response: Thank you for the keen comment. As your advice, we reconducted McNemar test to compare the nodule detection sensitivities of each radiologist and VCAR between DLIR-LDCT and DLIR-ULDCT and to compare the nodule detection sensitivity of VCAR and LuCAS. Each p value number itself has changed, but the overall statistical significance has not changed. The changes were as follows: 

Radiologists’ nodule detection sensitivities between DLIR-LDCT and DLIR-ULDCT, P=.57 → P=1

VCAR’s nodule detection sensitivity between DLIR-ULDCT and ASIR-V50-ULDCT, P=.99 → P=1

VCAR’s nodule detection sensitivity between DLIR-LDCT and DLIR-ULDCT, P=.64 → P=.5

Nodule detection sensitivity between VCAR and LuCAS, P=.113 → P=.18

Subsolid nodule detection sensitivity between VCAR and LuCAS, P=.002 → P=.03

7. One of the major limitations in this study is that the authors used a vendor-specific DLIR algorithm, and as a result, the applicability to other DLIR algorithms provided by different vendors is not discussed. Recent investigations employed a different vendor-specific DLR for ULDCT of the lung to improve image quality (doi: 10.1016/j.acra.2022.04.025 and doi: 10.2214/AJR.19.22680.). Please provide a concise discussion of whether your results align with these prior investigations to offer readers the wide applicability of study findings.

Response: We appreciate the comments. As you suggested, we added previous studies of DLIR in discussion section as follows: 

There are previous studies of a vendor-specific and vendor-agnostic DLIR to create high quality images of ULDCT [26, 27]. In a study by Goto et al., ULDCT with vendor-specific DLIR (AiCE, Canon Medical Systems) showed better nodule CNR and comparable subjective image quality assessment (noise in air, noise in soft-tissue, streak artifact, texture fineness, and overall quality) compared to LDCT with iterative reconstruction in the phantom study [26]. In a subjective evaluation, ULDCT with DLIR of 14 nonobese patients achieved a significantly higher preference by radiologists compared to ULDCT with iterative reconstruction [26]. Hata et al. proved a vendor-agnostic DLIR for noise reduction of ULDCT showed better subjective image quality compared to the ULDCT with iterative reconstruction [27]. Also, nodule detection rate of radiologists improved in ULDCT with DLIR compared to ULDCT with iterative reconstruction [27]. These previous studies and ours both align with the same result that ULDCT with DLIR increases objective and subjective image quality compared to iterative reconstruction algorithms and provides image quality comparable to that of LDCT.

8. The study findings may be applicable for pediatric CT, where radiation exposure is of concern in comparison to adults who require lung cancer screening. Discussing the applicability to low-dose pediatric CT by citing the relevant articles (such as doi: 10.1148/rg.2021210105. and doi: 10.1186/s12880-021-00677-2.) would further emphasize the value of the authors' findings.

Response: Dose reduction in pediatric CT using DLIR and ULDCT is a very novel and influential idea and thank you for suggesting a perspective that we hadn’t thought of before. We added this to discussion section as follows: 

There have been concerns regarding the increased radiation exposure from medical imaging of pediatric populations [28]. Efforts to reduce the radiation dose in pediatric CT imaging continued steadily and dose reduction strategies in CT imaging is expected to dramatically reduce the adverse effects of radiation exposure such as radiation-induced cancers [28, 29]. One of the proven dose reduction strategies is DLIR which can minimize image noise while preserving the adequate resolution of the CT scans [29, 30]. In a study by Yoon et al., in pediatric non-contrast and contrast-enhanced chest CT (CTDIvol, 1.3 ± 0.5 mGy; DLP 49.0 ± 26.3 mGy × cm; effective dose, 2.2 ± 3.2 mSv), DLIR showed significant noise reduction with increased SNR and CNR and showed better subjective overall image quality and noise compared to iterative reconstruction [30]. This result is comparable to our results and application of DLIR is expected to substantially reduce the radiation dose while maintaining the appropriate image quality for diagnosis in pediatric chest CT scans.

---

## [Decision Letter · Decision Letter 2]

4 Jan 2024

Prospective Evaluation of Deep Learning Image Reconstruction for Lung-RADS and Automatic Nodule Volumetry on Ultralow-Dose Chest CT

PONE-D-23-05687R2

Dear Dr. Yoon:

We’re pleased to inform you that your manuscript has been judged scientifically suitable for publication and will be formally accepted for publication once it meets all outstanding technical requirements.

Kind regards,

Gayle E. Woloschak, PhD

Section Editor

PLOS ONE

Additional Editor Comments (optional):

Thank you for addressing the concerns raised by the reviewers.

Reviewers' comments:

Reviewer's Responses to Questions

**Comments to the Author**

1. If the authors have adequately addressed your comments raised in a previous round of review and you feel that this manuscript is now acceptable for publication, you may indicate that here to bypass the “Comments to the Author” section, enter your conflict of interest statement in the “Confidential to Editor” section, and submit your "Accept" recommendation.

Reviewer #3: (No Response)

2. Is the manuscript technically sound, and do the data support the conclusions?

Reviewer #3: (No Response)

3. Has the statistical analysis been performed appropriately and rigorously? 

Reviewer #3: (No Response)

4. Have the authors made all data underlying the findings in their manuscript fully available?

Reviewer #3: (No Response)

5. Is the manuscript presented in an intelligible fashion and written in standard English?

Reviewer #3: (No Response)

6. Review Comments to the Author

Reviewer #3: (No Response)

7. PLOS authors have the option to publish the peer review history of their article (what does this mean?). If published, this will include your full peer review and any attached files.

Reviewer #3: No
